# Automated and Quantitative Mineralogy Applied to Chromite Ore Characterization and Beneficiation

Mark I. Pownceby *, David A. McCallum and Warren J. Bruckard

Commonwealth Scientific and Industrial Research Organisation (CSIRO) Mineral Resources, Private Bag 10, Clayton South, VIC 3169, Australia
* Correspondence: mark.pownceby@csiro.au

**Abstract:** A characterization study of chromite ore from South Africa was conducted using bulk assays, X-ray diffraction, optical, scanning electron microscopy (SEM), automated electron probe microanalysis (EPMA) and quantitative evaluation of mineral by scanning electron microscopy (QEMSCAN) mineralogical techniques, and quantitative EPMA. The aim was to identify all major gangue impurities, the degree of chromite liberation, and possible beneficiation options. The bulk material assayed 40.5% $Cr_2O_3$ with the major impurities being $Al_2O_3$ (13.2%), MgO (12.1%), and $SiO_2$ (7.5%). Quantitative mineral phase analysis showed that the sample mineralogy was dominated by a chrome-rich spinel phase with an average chemical composition (in wt.%) of: $Cr_2O_3$—47.8; FeO—26.0; $Al_2O_3$—15.4; and MgO—11.0. Contaminant phases included siliceous minerals enstatite, anorthite-rich plagioclase (bytownite), Cr-rich diopside (containing 1–2 wt.% $Cr_2O_3$), and phlogopite mica. QEMSCAN analysis of sized fractions indicated that (a) most silicate gangue species were in the +850 μm fractions, (b) the chrome-rich spinel in all fractions was >80% liberated, and (c) the most common mineral association for chromite was with enstatite. Based on the results, upgrading test work demonstrated that stage crushing followed by wet gravity concentration produced a chemical–metallurgical-grade 'chromite' product containing >46% $Cr_2O_3$ and <1% $SiO_2$.

**Keywords:** automated mineralogy; QEMSCAN; electron probe microanalysis; chromite





## 1. Background

Chromium is found in a wide variety of oxide and silicate minerals in the Earth's crust and mantle. Chromite is the only ore mineral from which metallic chromium and chromium compounds are obtained, with the most important use of chromium being in the production of high-strength alloys and alloys which are heat-, abrasion-, corrosion-, and oxidation-resistant. Approximately 90% of the world's production of 41 Mt tons (2022) of chromite is used for this purpose [1]. The remaining 10% of the production is consumed in refractory, chemical, and foundry industries. World resources are greater than 12 billion tons of shipping-grade chromite, sufficient to meet conceivable demand for centuries [1]. World chromium resources are heavily geographically concentrated (95%) in Kazakhstan and southern Africa.

Chromite can be classified on the Cr/Fe ratio and, according to its end use, is classified as metallurgical, chemical, or refractory grade. The highest-grade chromites are those with a Cr/Fe ratio of more than 2 and containing a minimum of 46 to 48% $Cr_2O_3$. Metallurgical-grade chromite used in alloy steel making contains 48% $Cr_2O_3$, and has a Cr:Fe ratio of 3:1 and less than 10% silica. Chemical-grade (high-iron) chromite used to manufacture sodium dichromate, and thereafter many different pigments, generally has a $Cr_2O_3$ content of >44%, a Cr:Fe ratio of >1.5:1, and a silica content of less than 3.5%. Refractory-grade chromite typically has a low $Cr_2O_3$ content (30%–40%), a Cr:Fe ratio between 2 and 2.5:1, and a relatively high (+25%–30%) alumina content [2,3]. The chromite used in refractories must have less than 3.5% silica, as well as limited silicates, such as pyroxenes and serpentine, which tend to form low melting phases.

Chromite occurs as a primary accessory mineral in basic and ultrabasic igneous rocks [4]. Economic deposits form via the crystallization of chromite from a cooling magma, resulting in large stratiform deposits or smaller pod-like deposits (podiform). The most significant occur in large, layered, and igneous intrusions in shield areas older than 2.06 billion years. The largest and best known of these is the stratiform Bushveld Complex (Republic of South Africa), which extends over 64,340 km$^2$ and contains over 60% of the world's known chromite reserves, and provides for more than 50% of the world's production [3,5].

Chromite ores are typically mixtures of chromite and gangue minerals, particularly orthopyroxene, olivine, and plagioclase, as well as hydrated alteration products such as talc, serpentine, and chlorite. Deposits that have been affected by metasomatism may also contain dolomite, magnesite, brucite, kaolin, quartz, and goethite. They therefore usually require some form of upgrading to produce a marketable product. The processing of chromite ore to produce chromite for refractory, chemical, and metallurgical markets usually involves crushing and grinding, as well as size sorting. Most chromite ores also require some degree of further beneficiation; this is mainly associated with gravity separation, but can also include magnetic separation or, in some cases, flotation [2,6].

The current study involved the mineralogical characterization of a South African chromite ore from the Ruighoek chromite deposit located in the western sector of the Bushveld Complex [7,8]. The aims of the study were to fully characterize the chromite ore using bulk assays, X-ray diffraction, and optical and automated electron beam microanalysis techniques, and to use the mineralogical data to provide input into the design of a beneficiation treatment process to upgrade the chromite to a marketable chemical–metallurgical-grade product containing >46% $Cr_2O_3$ and <1% $SiO_2$.

## 2. Nomenclature

The term chromite is used in many ways and the terminology is not consistent throughout industry and academia. The mineral chromite, *sensu stricto*, is the endmember composition $FeCr_2O_4$ within the spinel group, which has the general formula $AB_2O_4$, where A and B represent tetrahedrally coordinated divalent (Fe, Mg, Mn, Zn, and Ni) and octahedrally coordinated trivalent (Fe, Al, and Cr) metal ions, respectively. In addition, the substitution of $Ti^{4+}$ into the B site (as in the case of ulvöspinel—$Fe_2TiO_4$) can occur. This relies on a coupled substitution $2B^{3+} = Ti^{4+} + A^{2+}$ mechanism, relative to the general $AB_2O_4$ formula. Therefore, compositionally, spinel analyses can be complex with a range of possible element substitutions. In practice, however, spinels can usually be treated in terms of six important endmember components: $FeCr_2O_4$-$MgCr_2O_4$-$FeAl_2O_4$-$MgAl_2O_4$-$Fe_3O_4$-$MgFe_2O_4$, and, since most spinels form at high temperatures, compositions of the mineral chromite tend to form extensive solid solutions with other endmember minerals within the spinel group, leading to potentially complex chemistries.

## 3. Experimental

A ~20 kg sample of lumpy 'chromite' ore from South Africa (nominally 95% passing 100 mm in size) was used in the test work. The lump sample was crushed to −2 mm and then sized into twelve fractions for further analysis. Sample weights and chemical assay results for each of the size fractions are provided in Table 1.

The −2 mm bulk sample was examined by X-ray diffraction (XRD, Philips X'Pert Pro, Philips, Eindhoven, The Netherlands), while the −2 mm sample and selected sized fractions were analyzed using scanning electron microscopy (SEM, FEI Quanta 400F, FEI, Hillsboro, OR, USA), QEMSCAN (E340 QEMSCAN, FEI, Hillsboro, OR, USA), and electron probe microanalysis (EPMA, Model JXA-8900R, JEOL Ltd., Akishima, Tokyo, Japan) techniques.

Table 1. Chemical assay and mass data for the initial twelve size fractions, plus a list of size fractions that were combined and analyzed via QEMSCAN.

| Sample Size Fraction (μm) | Oxide (wt.%) | | | | | | | | | | Cumulative Cr₂O₃ (wt. %) | Mass Percent (−2mm bulk) | QEMSCAN Fractions (mass %) |
|---|---|---|---|---|---|---|---|---|---|---|---|---|---|
| | $SiO_2$ | $Al_2O_3$ | $Fe_2O_3$ | CaO | $Cr_2O_3$ | MgO | $Mn_3O_4$ | $Na_2O$ | $TiO_2$ | $V_2O_5$ | | | |
| +1700 | 17.3 | 10.9 | 21.4 | 0.873 | 32.5 | 16.1 | 0.202 | 0.22 | 0.412 | 0.240 | 2.16 | 2.7 | 16.7 |
| −1700 +1180 | 17.1 | 10.8 | 21.4 | 0.777 | 32.6 | 16.3 | 0.207 | 0.17 | 0.435 | 0.240 | 13.40 | 14.0 | |
| −1180 +850 | 15.9 | 11.1 | 21.9 | 0.695 | 33.7 | 15.9 | 0.200 | 0.18 | 0.431 | 0.251 | 22.70 | 11.2 | 11.2 |
| −850 +600 | 8.42 | 12.8 | 24.6 | 0.414 | 39.9 | 13.0 | 0.205 | 0.19 | 0.511 | 0.296 | 30.77 | 8.2 | 8.2 |
| −600 +425 | 3.69 | 13.9 | 26.5 | 0.261 | 44.1 | 11.2 | 0.210 | 0.20 | 0.546 | 0.319 | 43.48 | 11.7 | 11.7 |
| −425 +300 | 2.11 | 14.2 | 27.2 | 0.225 | 45.4 | 10.5 | 0.208 | 0.21 | 0.550 | 0.331 | 56.70 | 11.8 | 26.2 |
| −300 +212 | 1.91 | 14.2 | 27.3 | 0.273 | 45.5 | 10.1 | 0.216 | 0.23 | 0.557 | 0.330 | 72.85 | 14.4 | |
| −212 +150 | 2.68 | 14.2 | 27.1 | 0.440 | 44.9 | 10.1 | 0.215 | 0.25 | 0.544 | 0.325 | 84.36 | 10.4 | 17.8 |
| −150 +106 | 4.49 | 14.3 | 26.4 | 0.749 | 43.4 | 10.4 | 0.199 | 0.27 | 0.543 | 0.313 | 92.27 | 7.4 | |
| −106 +75 | 7.55 | 14.2 | 24.6 | 1.24 | 40.2 | 10.8 | 0.196 | 0.35 | 0.522 | 0.298 | 97.13 | 4.9 | n.e. [†] |
| −75 +53 | 11.4 | 14.1 | 22.7 | 1.83 | 36.6 | 11.6 | 0.189 | 0.44 | 0.494 | 0.270 | 99.11 | 2.2 | n.e. |
| −53 | 15.9 | 13.5 | 20.7 | 2.29 | 32.7 | 12.6 | 0.192 | 0.46 | 0.537 | 0.245 | 100.00 | 1.1 | n.e. |
| Head Assay | 7.50 | 13.2 | 25.3 | 0.575 | 40.5 | 12.1 | 0.218 | 0.22 | 0.525 | 0.295 | | | |

[†] n.e. = not examined.

### 3.1. XRD, SEM, and EPMA Sample Preparation

The −2 mm bulk fraction was ground to a powder (grainsize < 1–2 μm) and then pressed flat into a standard aluminum sample holder in preparation for XRD analysis. For characterization with SEM and EPMA, grains from a split sub-sample of the unground −2 mm bulk and the −600 +425 μm fraction were uniformly dispersed in epoxy resin mounts, which were then polished flat using successively finer diamond pastes down to a final diamond paste cutting size of 1 μm. Prior to analysis, the sample mounts were coated with a 25 nm-thick carbon film to prevent charge build-up on the surface.

### 3.2. QEMSCAN Sample Preparation

Preliminary X-ray fluorescence (XRF, Bruker S8 Tiger, Bruker, Billerica, MA, USA) analysis of the twelve sized fractions indicated that approximately 90% of the chromia was present in the size range of −1700 to +106 μm. To reduce the number of samples required for analysis by QEMSCAN, selected size fractions were combined. In addition, the three finest size fractions were not analyzed due to the small contribution that they made to the overall chromium content. This gave six size fractions in total to be analyzed via QEMSCAN (Table 2).

**Table 2.** The number of blocks examined and particles analyzed using automated QEMSCAN mineralogical techniques.

| Size Fraction (μm) | Number of Blocks | Number of Particles |
|---|---|---|
| +1180 | 3 | 863 |
| −1180 +850 | 3 | 1573 |
| −850 +600 | 1 | 141 |
| −600 +425 | 1 | 533 |
| −425 +212 | 1 | 1906 |
| −212 +106 | 1 | 2421 |

A representative ~0.4 g sample of the sized samples was mixed with high-purity graphite, approximately equal to the sample weight, to help with particle separation and to reduce particle agglomeration. Each sample mixture was added to ~20 g of epoxy resin and stirred to ensure the complete wetting of all the particles, before then being left overnight in a high-pressure chamber to compress any air bubbles. After curing, the sample blocks were polished to create a smooth analysis surface, before coating with a 25 nm-thick carbon film.

### 3.3. Phase Characerization by XRD

Qualitative XRD analysis of the −2 mm bulk ore was undertaken to identify the major crystalline phases that were present. An XRD pattern was collected with a Philips X'Pert Pro diffractometer (Philips, Eindhoven, The Netherlands), operating with Bragg–Brentano configuration and using copper Kα radiation. The incident beam consisted of a 0.04 rad Soller slit and a 1° fixed divergence slit. The divergent beam consisted of a 0.3 mm receiving slit, a 0.04 rad Soller slit, a 1° anti-scatter slit, a curved graphite monochromator, and a PW1711 proportional detector. The data were collected from 3.5 to 80° 2θ with a step size of 0.02° and a collection time of 0.4 s per step. The phases were identified by comparing the peak positions and intensities with the data published by the International Centre for Diffraction Data (ICDD).

### 3.4. Phase Characerization by SEM

SEM imaging of the −2 mm bulk and the −600 +425 μm fraction was conducted under high-vacuum conditions in a FEI Quanta 400F field emission environmental scanning electron microscope (FEG-ESEM, FEI, Hillsboro, OR, USA). Backscattered electron (BSE) images were taken using an accelerating voltage of 20 kV and a probe current of approximately 150 pA.

The identity of phases was qualitatively confirmed using an EDAX energy-dispersive X-ray analysis system (AMETEK Inc., Berwyn, PA, USA). The spectra were collected for 60 s and element peaks were identified using EDAX Genesis software.

### 3.5. Electron Probe Microanalysis (EPMA)

The −2 mm bulk and the −600 +425 μm fraction were examined by EPMA using a combination of high-resolution mapping and wavelength-dispersive (WD) quantitative elemental analysis. All analyses were performed using a JEOL Superprobe microanalyzer (Model JXA-8900R, JEOL Ltd., Akishima, Tokyo, Japan).

### 3.5.1. EPMA Mapping

A series of maps showing the distribution of the elements Fe, Ca, Na, Mn, Al, Cr, Ti, Mg, Zn, and Si were collected for both samples. These ten elements were selected for analysis based on (a) the XRF analysis of the bulk −2 mm sample, which indicated that these elements were all >0.2 wt% in abundance (except for Zn, which was generally <0.1 wt%), and (b) the fact that these elements were considered the minimum number required to delineate the major phases identified by XRD results. The previous sample inspection via optical microscopy and BSE imaging indicated that the sample was mineralogically and texturally complex, and therefore areas selected for mapping covered regions that were considered representative of the dominant textural and mineralogical types found to be present.

Operating conditions for the microprobe were as follows: an accelerating voltage of 20 kV, a beam current of 100–110 nA, a step size of 5 μm, and counting times of 20 ms per step. The map for the −2 mm bulk sample covered a region spanning 4000 μm × 4000 μm (i.e., 800 × 800 steps at a step size of 5 μm), whilst the area mapped for the −600 +425 μm sample covered 5120 μm × 5120 μm. The choice of step size was based on a compromise between maximizing the number of particles analyzed and ensuring that any fine-grained mineral phases were located. Standards used for the mapping analysis were: spinel ("Magalox", $MgAl_2O_4$), hematite ($Fe_2O_3$), halite (NaCl), zinc sulphide (ZnS), eskolaite ($Cr_2O_3$), MnFe alloy, rutile ($TiO_2$), and wollastonite ($CaSiO_3$).

Elements that were not measured by WD spectroscopy were measured with two energy-dispersive (ED) spectrometers operating in parallel. Measuring both ED and WD signals simultaneously ensured that the complete chemical spectrum was obtained at each step interval in the map. This additional information was important when trying to identify the phases that contained elements that were not found in the main WD element map suite.

After mapping, the element distribution data were manipulated using the software package 'CHIMAGE' (v. 10.0.34, CSIRO, Clayton, VIC, Australia) [9]. CHIMAGE allows the individual element data to be displayed as either scatter plots or as combined element maps (where the data for two or more elements are combined on the one mapped region), thereby making any correlations between elements readily identifiable [10]. Phase identification and modal analysis were performed using the 'Opal' phase analysis library and methods proposed by Torpy et al. [11]. Mineral species were identified based on X-ray spectrum matching and comparison with compositions of known mineral species.

### 3.5.2. Quantitative EPMA

Quantitative electron probe microanalyses were performed in the wavelength-dispersive (WD) mode. The electron probe was operated at an accelerating voltage of 20 kV and a beam current of 100 nA. Approximately 5–10 randomly targeted spot analyses of each of the phases identified via the mapping procedure were examined and their compositions were averaged. At each analysis position, the following suite of elements was quantitatively measured: Fe, Ca, Na, Al, Cr, K, Ti, Mg, Zn, and Si. Oxygen was calculated by difference, based on valence. Standards used were spinel ("Magalox", $MgAl_2O_4$), hematite ($Fe_2O_3$), halite (NaCl), zinc sulfide (ZnS), eskolaite ($Cr_2O_3$), rutile ($TiO_2$), and wollastonite ($CaSiO_3$).

## 3.6. QEMSCAN Analysis

QEMSCAN analysis was completed using an E340 QEMSCAN (FEI, Hillsboro, OR, USA), equipped with four X275HR silicon drift detectors (SDDs). Samples were analyzed via the X-ray mapping of each fraction, and mineral species were automatically inferred from the chemical signature from the characteristic X-ray spectra from each spot analysis. Data analysis was conducted using iDiscover™ software for QEMSCAN. Multiple blocks of the coarser size fractions were analyzed to provide enough particles to provide a statistically valid result (Table 2).

## 4. Results and Discussion

### 4.1. Bulk Chemistry

Chemical analysis by XRF indicated that the bulk ore had a $Cr_2O_3$ content of 40.5 wt.% and a $SiO_2$ content of 7.5 wt.% (Table 1). The sample also contained significant levels of $Al_2O_3$ (13.2 wt.%) and $MgO$ (12.1 wt.%), suggesting significant replacement in the spinel structure of Cr by Al and $Fe^{2+}$ by Mg. Minor impurities included calcium (0.58 wt.%), manganese (0.22 wt.%), sodium (0.22 wt.%), titanium (0.53 wt.%), and vanadium (0.30 wt.%). Calcium and sodium are most likely associated with gangue mineral grains, such as feldspars and pyroxenes; however, manganese, titanium, and vanadium can all have limited solid substitution in the spinel structure.

Individually sized fractions were analyzed by XRF to determine any variation in bulk chemistry based on size, and the assay data were combined with the sizing data to give a cumulative elemental distribution across the size fractions. The distribution data for the two oxides of interest, $Cr_2O_3$ and $SiO_2$, are shown graphically in Figure 1. The results indicate that a significant portion of the siliceous material was present in the +600 μm fractions of the −2 mm material with ~70% of the total $SiO_2$ present in the top four size fractions (i.e., +1700, −1700 +1180, −1180 +850, and −850 +600 μm fractions). In comparison, only 31% of the total $Cr_2O_3$ was present in these size ranges. This suggests that, initially, some of the high silica gangue could be removed by a simple screening procedure without too much of a loss of $Cr_2O_3$.

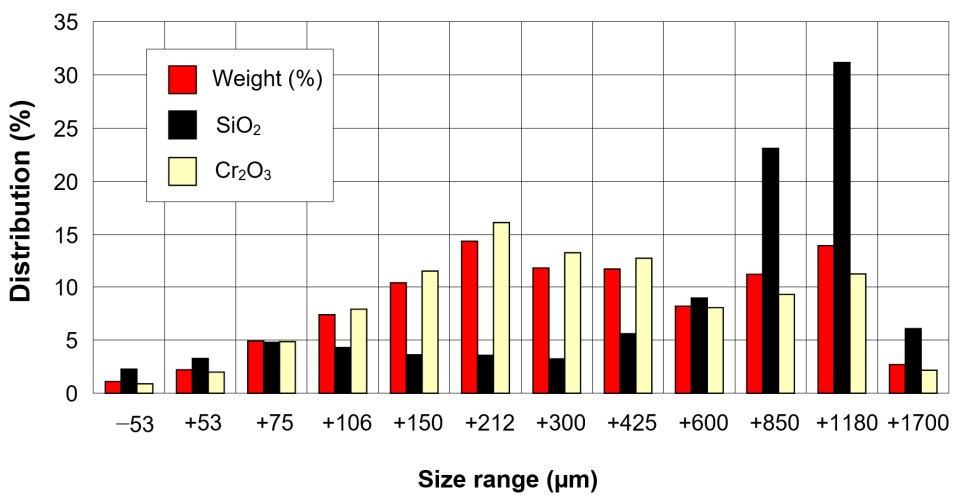

**Figure 1.** $SiO_2$ and $Cr_2O_3$ distribution by size for the −2 mm bulk chromite sample.

### 4.2. Bulk Mineralogy—XRD Analysis

Characterization results from the XRD analysis of the −2 mm bulk sample showed that the mineralogy was dominated by the spinel phases ferrian–magnesiochromite ([FeMg]$Cr_2O_4$) and aluminum–chromite (Fe[AlCr]$_2O_4$), with enstatite- ($Mg_2Si_2O_6$), hematite- ($Fe_2O_3$), and anorthite-rich plagioclase ($CaAl_2Si_2O_8$) as the accessory phases. Although the sample was identified as a 'chromite' ore, chromite, i.e., pure $FeCr_2O_4$, was not identified as a unique phase in this sample. All of the chromium present in the sample was contained within the

spinel phases ferrian–magnesiochromite and aluminum–chromite—these represent solid solution compositions between the pure $FeCr_2O_4$ (chromite), $FeAl_2O_4$ (hercynite), $MgCr_2O_4$ (magnesiochromite), and $MgAl_2O_4$ (spinel) endmembers. Since the term 'chromite' strictly refers to the FeCr-rich endmember and the grains in this sample are a mixture between different spinel types, in the following sections, the term 'chromite' is replaced by the more generic term 'spinel'.

*4.3. Scanning Electron Microscopy*

Representative BSE images from the two fractions examined by SEM are provided in Figure 2. The individual mineral phases shown were inferred based on the chemistry provided via EDAX X-ray spectrum analysis.

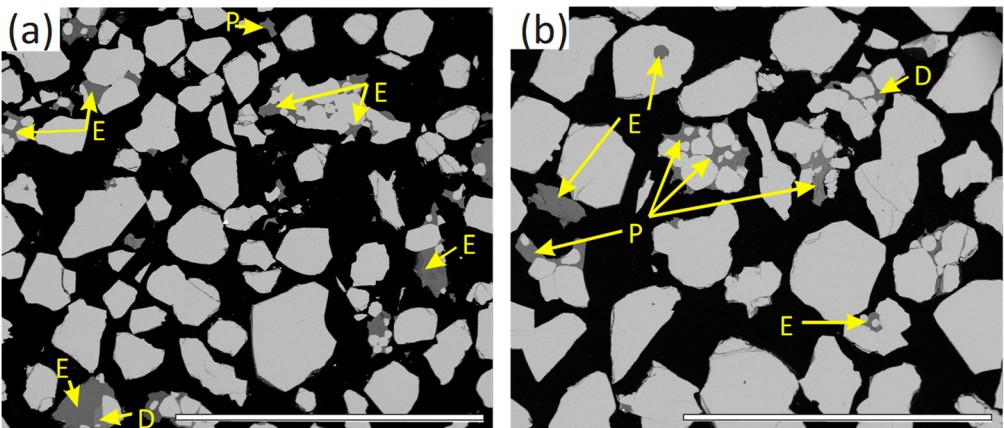

**Figure 2.** BSE images of (**a**) the −2 mm bulk sample and (**b**) the −600 +425 μm fraction. Key to symbols: E = enstatite, P = plagioclase (anorthite-rich), and D = diopside. The bright grains in both images are chromite-rich spinels. The scale bar in each image represents a size of 2.0 mm.

The mineral textures exhibited by the samples were similar, comprising a coarse-grained assemblage characterized by irregular-shaped, optically homogeneous, mostly liberated spinel grains. In addition to the liberated spinel grains, there was a smaller proportion of (a) 'composite' particles, consisting of spinel and silicate mineral phases bonded together; (b) fully liberated silicate grains; and (c) some larger spinel grains containing inclusions of silicates (e.g., the rounded enstatite inclusion in the spinel grain towards the top LHS shown in Figure 2b). Qualitative EDAX analysis of the spinel grains (both liberated and composite types) suggested a relatively unvarying composition dominated by the elements Cr, Mg, Al, and Fe. This composition implied that the spinel grains were part of a continuous solid solution among the endmembers $FeCr_2O_4$-$MgCr_2O_4$-$FeAl_2O_4$-$MgAl_2O_4$, consistent with the XRD results. We note, however, that there may also be some $Fe^{3+}$ present in the chrome-rich spinel grains (as the oxidized magnetite spinel component, $Fe_3O_4$), although it was not possible to confirm this using a microbeam-based analysis technique.

Analyzing the silicate material which 'locks' the chrome-rich spinel grains together to form the composite particles indicated a range of mineral types. The most common silicate phase was a Mg-rich silicate, which also contained some minor Fe. Based on the XRD examination, this mineral was most likely the orthorhombic pyroxene, enstatite ($Mg_2Si_2O_6$). A second silicate mineral containing Ca, Al, and minor Na was also identified in the composite particles. The Al:Si ratio of this mineral suggested it was most likely a Ca-rich plagioclase feldspar. This conclusion is consistent with the XRD results, which indicated the presence of anorthite-rich plagioclase ($CaAl_2Si_2O_8$) in the sample. Very small occurrences of a second pyroxene phase were also observed during the SEM examination. Compared to the enstatite, this mineral was characterized by lower levels of Fe, less Mg, and a higher proportion of Ca. This mineral was present at an abundance which was below the detection limit achievable using XRD (~1%–2%), so a definitive identification was not

possible; however, based on its chemistry, it is most likely the monoclinic pyroxene, known as diopside (Ca(Mg,Fe)[$Si_2O_6$]).

### 4.4. Quantitative Elctron Probe Microanalysis

Quantitative EPMA results for chrome-rich spinel, anorthite-rich plagioclase, enstatite, diopside, and mica showed that these minerals were all homogeneous in their chemical compositions. A small number of other unidentified minerals were also evident in the sample. Based on their chemistry, various types of aluminosilicate alteration phases were observed, possibly clays and clay-like minerals, as well as minor amounts of an Fe-rich phase(s)—possibly hematite ($Fe_2O_3$) or goethite (FeO.OH).

The average EPMA data for the main minerals are provided in Table 3. For the chrome-rich spinel phase, the compositions of all grains examined from both samples were grouped together and averaged. This gave average chrome-rich spinel compositions (in wt.%) of $Cr_2O_3$: 47.8 (0.4), FeO: 26.0 (0.6), $Al_2O_3$: 15.4 (0.5), and MgO: 11.0 (0.4). The averaged values were only calculated for the elements normally present in spinel (although spinel may also contain minor amounts of other impurities such as $TiO_2$, ZnO, and MnO etc.). The values in brackets indicate one standard deviation (1 σ).

**Table 3.** The average quantitative EPMA data for key minerals in the −2 mm bulk sample and the −600 +425 μm fraction.

| Sample | Phase | Oxide (wt. %) | | | | | | | | | Total [†] |
|---|---|---|---|---|---|---|---|---|---|---|---|
| | | $SiO_2$ | FeO | CaO | $Na_2O$ | $Al_2O_3$ | $Cr_2O_3$ | $TiO_2$ | MgO | ZnO | |
| −2 mm bulk sample | | | | | | | | | | | |
| | Spinel (Cr) | 0.01 | 25.93 | 0.01 | 0.01 | 15.46 | 47.78 | 0.71 | 10.89 | 0.10 | 100.90 (24) |
| | Anorthite | 50.54 | 0.23 | 13.55 | 3.60 | 31.42 | 0.33 | 0.01 | 0.01 | 0.01 | 99.70 (7) |
| | Enstatite | 56.03 | 7.34 | 0.86 | 0.01 | 1.36 | 0.49 | 0.10 | 35.40 | 0.00 | 101.59 (13) |
| | Diopside | 53.04 | 3.07 | 23.01 | 0.40 | 1.81 | 1.60 | 0.20 | 17.64 | 0.01 | 100.79 (2) |
| | Mica | 38.93 | 2.92 | 0.01 | 0.62 | 15.45 | 1.65 | 4.66 | 23.16 | 0.00 | 87.41 (2) [‡] |
| −600 +425 μm fraction | | | | | | | | | | | |
| | Spinel (Cr) | 0.01 | 26.16 | 0.00 | 0.00 | 15.42 | 47.92 | 0.60 | 11.04 | 0.10 | 101.25 (19) |
| | Anorthite | 49.31 | 0.21 | 14.40 | 3.09 | 31.30 | 0.20 | 0.01 | 0.01 | 0.01 | 98.54 (10) |
| | Enstatite | 55.84 | 7.19 | 1.08 | 0.02 | 1.34 | 0.51 | 0.10 | 33.63 | 0.01 | 99.70 (19) |
| | Diopside | 52.73 | 3.05 | 22.76 | 0.42 | 2.00 | 1.07 | 0.18 | 16.70 | 0.01 | 98.92 (6) |
| | Mica | 39.51 | 2.51 | 0.00 | 0.71 | 15.02 | 1.69 | 4.41 | 22.97 | 0.02 | 86.85 (4) [‡] |

[†] The numbers in brackets indicate the number of analyses averaged for each mineral phase. [‡] Low totals for the micaceous phase are due to the presence of water, as well as elements not included in the analysis dataset (e.g., potassium and fluorine).

Using the averaged data, and based on the typical spinel compositional formula $AB_2O_4$, where A = divalent cations ($Fe^{2+}$, Mg, and B) = trivalent cations ($Fe^{3+}$, Al, and Cr), the spinel present in the 'chromite' sample had the following calculated structural formula: $(Fe^{2+}_{0.5}, Mg_{0.5})(Fe^{3+}_{0.2}, Al_{0.6}, Cr_{1.2})_2O_4$. Note that the EPMA results indicate excess iron in the sample, leading to some of the Fe being present in the $Fe^{3+}$ state. The structural formula was therefore calculated based on known spinel stoichiometry, using the method described by Droop [12].

For the other phases present, the plagioclase was part of the albite–anorthite solid solution series, Na[$AlSi_3O_8$]-Ca[$Al_2Si_2O_8$], with a composition consistent with that of bytownite (i.e., between 70 and 90 mol% of the $CaAl_2Si_2O_8$ component). Minor Fe and Cr (replacing Al) were also present in the plagioclase. The pyroxene phases, enstatite and diopside, all showed the typical range in major and minor elements commonly analyzed in these minerals [13]. It was noted, however, that the relatively high $Cr_2O_3$ content present in the diopside grains (~1–2 wt.%) suggested it may be best classified as chromium-rich diopside.

In addition to the major pyroxene and plagioclase silicate phases, the EPMA data indicated the presence of an additional minor aluminum–silicate mineral. Owing to the

absence of key elements in the analysis dataset, complete quantitative results were not available for this mineral; however, the measured Al:Si ratio of ~1:3 was consistent with this material being micaceous. Furthermore, based on the analyzed FeO and MgO contents, it is likely that this mineral was phlogopite mica, $K_2(Mg,Fe^{2+})_6[Si_6Al_2O_{20}](OH,F)_4$.

### 4.5. EPMA Mapping

Results from the EPMA mapping are shown in Figure 3 for the −2 mm bulk sample where a back-scattered electron (BSE) image, together with selected key element concentration maps, is illustrated. Regions of high element correlation between individual maps identify the locations and compositions (qualitative) of the different mineral phases present. For example, enstatite grains are indicated by white regions (=high concentration) on the Mg and Si distribution maps, while white regions on the Al and Ca distribution maps indicate plagioclase (the highest Al- and Ca-containing mineral in the sample). Figure 3 also shows a composite element map which overlays the concentration data for the elements Fe, Cr, and Si. In this map, each individual element was assigned a primary color, e.g., Fe = blue, Cr = green, and Si = red. Using this color scheme, chrome-rich spinel grains are light blue in color (a mixture of Fe and Cr), while gangue silicate grains are red in color, indicating that they were low in Cr (green) and Fe (blue).

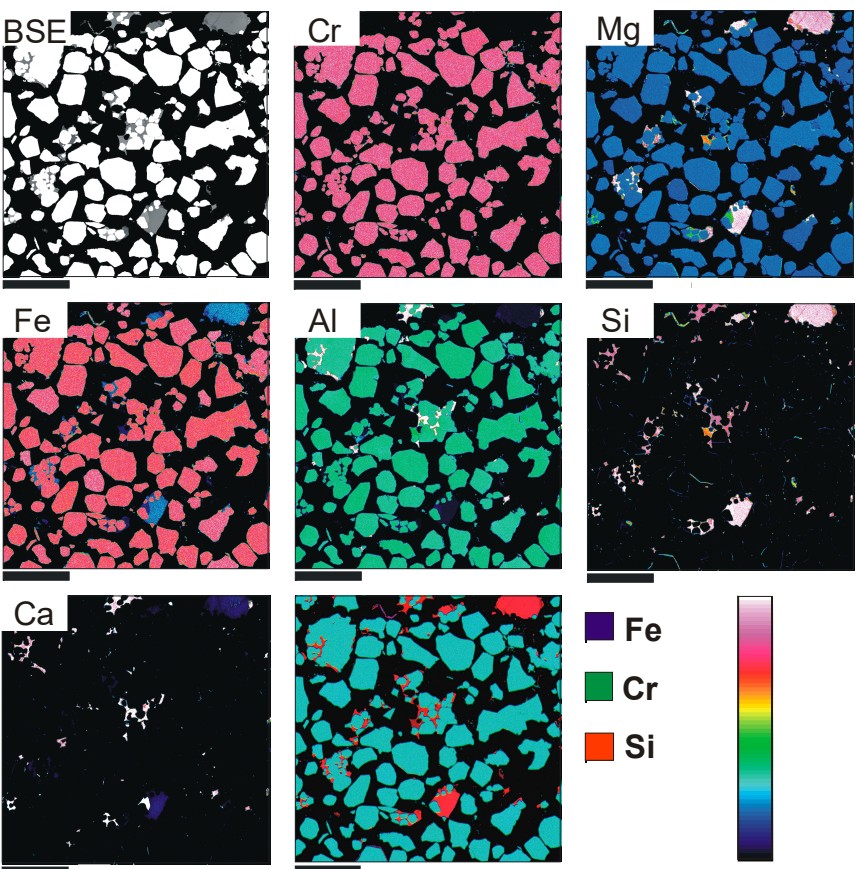

**Figure 3.** Back-scattered electron (BSE) selected individual element concentration maps and a composite map for the −2 mm bulk sample. The scale bar beneath each image indicates a size of 1.0 mm, while the colored bar at bottom right indicates the relative concentration of each element.

From an analysis of both the individual element concentration maps, together with the composite map and the selected element versus element scatter plots (not shown), it was possible to ascribe mineral names to the various grains to produce a classified mineral distribution map for each sample. The classified maps for the −2 mm bulk sample and the −600 +425 μm fraction are shown in Figure 4. These maps are particularly useful for

illustrating the relationships between the texture and mineralogy of the different phases present. For example, the results show that, texturally, the gangue minerals (plagioclase and diopside) were generally much smaller in grainsize than the chrome-rich spinel, and both occurred either intergrown with the chrome-rich spinel or as minor phases adhering to the liberated chrome-rich spinel grains. In contrast, the enstatite could be much more coarsely grained and was commonly found as liberated grains.

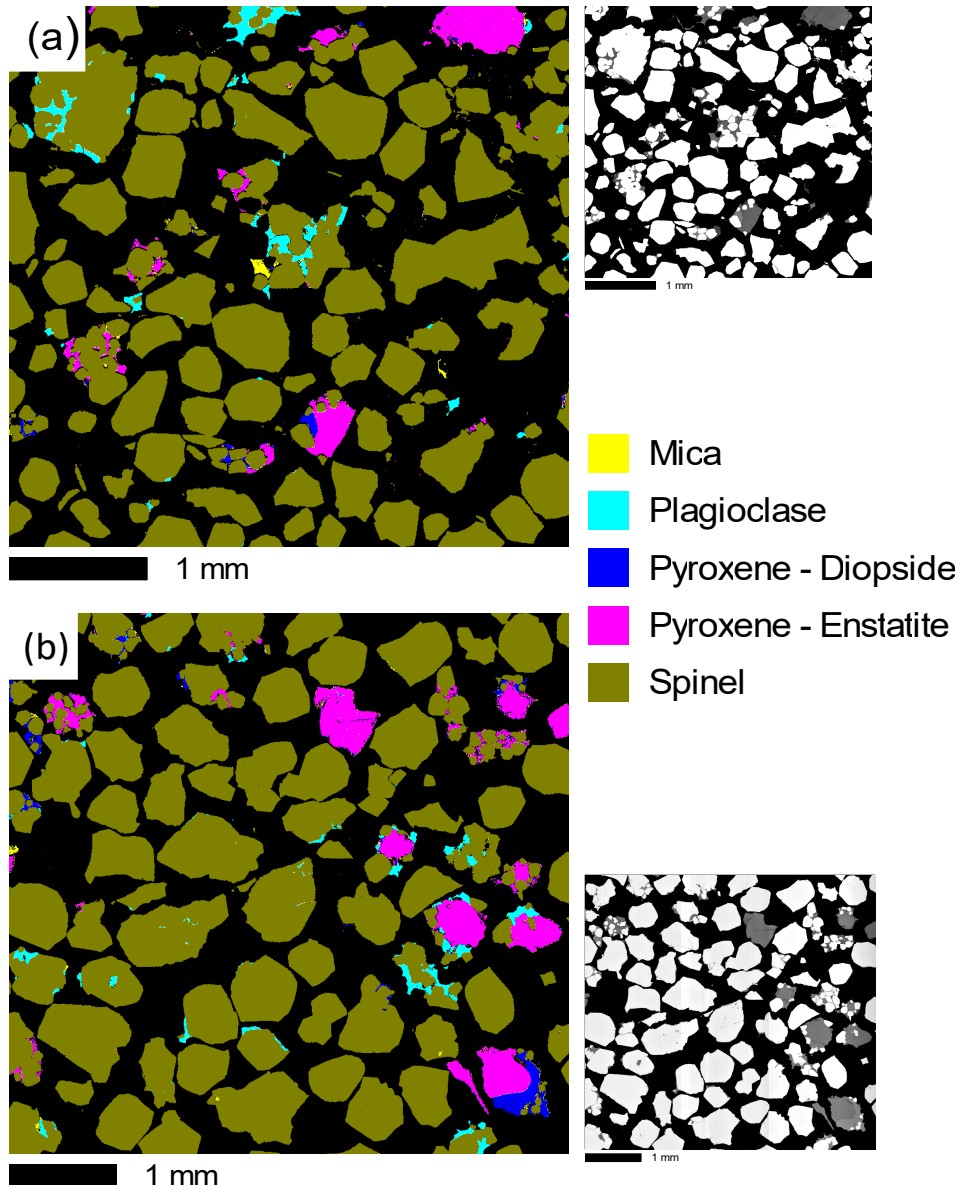

**Figure 4.** Phase-patched maps for (**a**) the −2 mm bulk sample and (**b**) the −600 +425 µm fraction. The BSE image for each of the mapped areas was included for comparison purposes.

The classified mineral maps may also be used to determine the proportions of the individual minerals. This can be achieved by an area analysis of each mineral species within the mapped areas. The results for this type of analysis are provided in Table 4 and show that both samples were similar in terms of the abundance of each identified mineral. Although this type of analysis is useful for comparing relative differences in modal mineralogy between samples, these values were measured in terms of area % (measured on the 2D mapped region), and therefore gave no information regarding the absolute mass percentages of individual phases or any information regarding the liberation of grains. Mass percentage and liberation data were determined using automated QEMSCAN analysis.

**Table 4.** Modal mineralogy data (in area %) from the EPMA map data.

| Mineral Phase | Abundance (Area %) | |
| :---: | :---: | :---: |
| | −2 mm Bulk Sample | −600 +425 μm Fraction |
| Spinel (Cr-rich) | 92.1 | 89.7 |
| Pyroxene–Enstatite | 4.5 | 6.9 |
| Pyroxene–Diopside | 0.5 | 1.3 |
| Anorthite | 2.5 | 1.7 |
| Mica | 0.4 | 0.4 |

*4.6. QEMSCAN Analysis*

4.6.1. Modal Mineralogy

QEMSCAN analysis provided quantitative modal mineralogical data for the six size fractions. The main minerals identified in the samples were chrome-rich spinel, enstatite, anorthite-rich plagioclase, hematite, and phlogopite. There was also a small amount of unidentified material, possibly representing the small amount of diopside and/or clay-like aluminosilicate minerals previously identified from the EPMA mapping and quantitative analyses. Modal mineralogy results for the minerals are provided in Table 5 in the form of the mineral area percentage and mineral mass percentage data.

**Table 5.** Modal mineralogy determined by QEMSCAN analysis.

| Parameter | Mineral | Size Fraction (μm) | | | | | | Total |
| :---: | :---: | :---: | :---: | :---: | :---: | :---: | :---: | :---: |
| | | +1180 | −1180 +850 | −850 +600 | −600 +425 | −425 +212 | −212 +106 | |
| | Spinel (Cr) | 64.18 | 58.69 | 77.95 | 89.78 | 93.02 | 76.66 | - |
| | Enstatite | 31.55 | 37.29 | 18.76 | 7.93 | 4.52 | 10.72 | - |
| Mineral | Anorthite | 3.80 | 3.50 | 2.73 | 1.63 | 1.69 | 10.67 | - |
| area (%) | Hematite | 0.06 | 0.05 | 0.01 | 0.01 | 0.01 | 0.02 | - |
| | Phlogopite | 0.30 | 0.16 | 0.11 | 0.09 | 0.08 | 0.39 | - |
| | Others | 0.11 | 0.31 | 0.44 | 0.56 | 0.68 | 1.54 | - |
| | Spinel (Cr) | 12.22 | 7.65 | 6.92 | 10.90 | 25.02 | 15.00 | 77.70 |
| | Enstatite | 4.01 | 3.25 | 1.11 | 0.64 | 0.81 | 1.40 | 11.23 |
| Mineral | Anorthite | 0.41 | 0.26 | 0.14 | 0.11 | 0.26 | 1.19 | 2.37 |
| mass (%) | Hematite | 0.01 | 0.01 | 0.00 | 0.00 | 0.00 | 0.00 | 0.03 |
| | Phlogopite | 0.03 | 0.01 | 0.01 | 0.01 | 0.01 | 0.04 | 0.12 |
| | Others | 0.01 | 0.02 | 0.02 | 0.04 | 0.10 | 0.16 | 0.36 |
| *Mass Percentages* | | *16.70* | *11.20* | *8.20* | *11.70* | *26.20* | *17.80* | *91.8* |

The mineral mass percentage data in Table 5 show that 77.7% of the chrome-rich spinel in the ore was present in the six size fractions coarser than 106 μm. These six fractions also contained significant amounts of the main silicate gangue minerals, enstatite and anorthite-rich plagioclase, which were present at levels of 11% and 2.5%, respectively. Data inspection for the individual fractions shows that the spinel grains were concentrated towards the mid-to-smaller-size ranges with the highest chrome-rich spinel levels (25%) occurring in the −425 +212 μm fraction. In comparison, the silicates tended to preferentially report to the coarse +850 μm fractions, reaching 30%–40% of the total mineral content in these fractions. The mineral area percentage results also show an increase in silicate gangue content in the coarser fractions, while the chrome-rich spinel was concentrated in the mid-to-smaller-size fractions. The QEMSCAN results are in excellent agreement with the assay data provided in Table 1, which shows that the bulk of the chromia was present in the mid-to-smaller-size fractions −850 +106 μm, whereas silica was highest in the coarse +850 μm fractions.

4.6.2. Particle Analysis and Mineral Liberation

Particle mineral maps for each size fraction confirmed the large amount of chrome-rich spinel present in the sample and provided a visual representation on a grain-by-grain basis

of the association between the chrome-rich spinel and gangue mineral grains. For example, the particle mineral maps measured for the +1180 μm and −600 +425 μm fractions are shown in Figure 5.

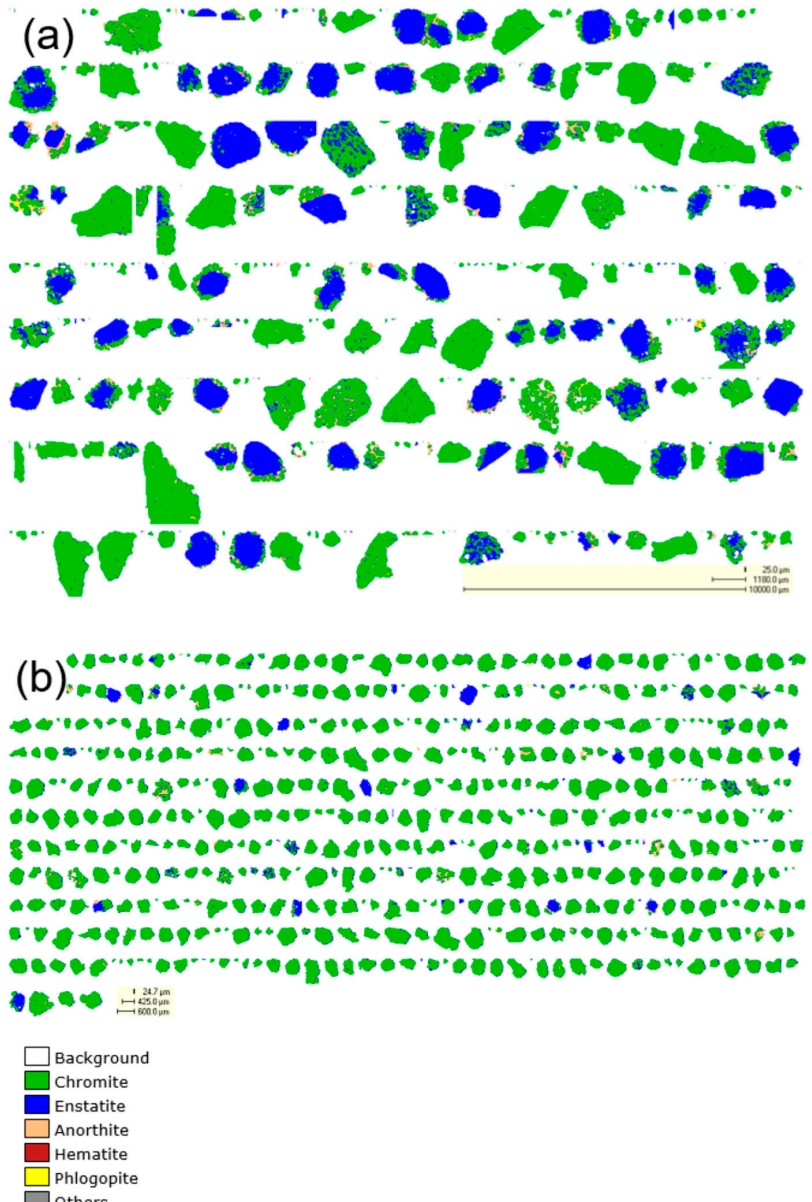

**Figure 5.** Particle mineral map from (**a**) the −2 mm bulk sample and (**b**) the −600 +425 μm fraction.

An examination of the particle maps for all six size fractions showed the following:

- the chrome-rich spinel was present in two main textural forms, either as fully liberated grains or as composite grains, where the chrome-rich spinel was intergrown with the gangue.
- the main spinel (Cr)–gangue association was between the spinel and the enstatite (which was the dominant gangue mineral); however, anorthite-rich plagioclase, phlogopite, and minor hematite were also found as inclusions within spinel grains or within the composite grains.
- enstatite was common either as fully liberated grains, as partly liberated grains with chrome-rich spinel adhering as thin coatings on the surface, or as composite spinel-enstatite particles.

- anorthite, phlogopite, and hematite were not found as fully liberated grains, except in the finest fractions examined.
- the abundance of anorthite-rich plagioclase increased with decreasing particle size.
- there was a significant increase in the ratio of spinel (Cr)–gangue toward the mid-to-smaller-size fractions (i.e., the fractions between −600 +106 μm).

The data collected on individual particles also facilitated the liberation and mineral association analysis for each phase. Liberation results for the chromite grains are given in Table 6 and are shown graphically in Figure 6. A summary of the mineral association values for the chrome-rich spinel is provided in Table 7.

The data in Table 6 and Figure 6 show the percentages of the chrome-rich spinel grain exposed to the background and other phases, relative to the total surface area of the chrome-rich spinel grain. Much of the chrome-rich spinel in all fractions was >80% liberated and the degree of liberation decreased as the grainsize increased. The mineral association data confirmed that the chrome-rich spinel became more liberated with decreasing grainsize. The main mineral association of chrome-rich spinel, in all size fractions, was with enstatite. This result is in good agreement with the visual observations taken from the EPMA mineral maps and the QEMSCAN particle maps.

**Table 6.** Spinel liberation data from the QEMSCAN analysis.

| Size Fraction (μm) | Cr Spinel Liberation | | | | | | | | | |
|---|---|---|---|---|---|---|---|---|---|---|
| | ≤10% | ≤20% | ≤30% | ≤40% | ≤50% | ≤60% | ≤70% | ≤80% | ≤90% | ≤100% |
| −212 +106 | 0.09 | 0.32 | 0.47 | 0.79 | 1.60 | 1.11 | 1.62 | 3.51 | 18.88 | 71.62 |
| −425 +212 | 0.01 | 0.01 | 0.04 | 0.15 | 0.14 | 0.40 | 0.59 | 1.33 | 5.00 | 92.33 |
| −600 +425 | 0.03 | 0.20 | 0.12 | 0.17 | 1.03 | 0.53 | 0.72 | 2.66 | 6.49 | 88.05 |
| −850 +600 | 0.20 | 0.27 | 2.36 | 1.88 | 1.70 | 0.77 | 0.75 | 3.91 | 10.10 | 78.06 |
| −1180 +850 | 0.95 | 2.89 | 4.43 | 4.30 | 3.52 | 4.22 | 3.05 | 4.15 | 8.55 | 63.95 |
| +1180 | 0.67 | 2.33 | 3.65 | 3.20 | 2.52 | 3.54 | 3.98 | 4.00 | 8.83 | 67.27 |

**Table 7.** Spinel association with other mineral phases (data from QEMSCAN analysis).

| Phase Association (%) | Size Fraction (μm) | | | | | |
|---|---|---|---|---|---|---|
| | +1180 | −1180 +850 | −850 +600 | −600 +425 | +425 −212 | −212 +106 |
| Background | 48.00 | 52.19 | 52.18 | 64.11 | 69.87 | 73.28 |
| Enstatite | 34.87 | 33.53 | 33.19 | 27.20 | 22.67 | 17.52 |
| Anorthite | 15.19 | 11.34 | 10.73 | 4.90 | 2.93 | 3.59 |
| Hematite | 0.23 | 0.23 | 0.04 | 0.03 | 0.01 | 0.01 |
| Phlogopite | 0.80 | 0.49 | 0.54 | 0.33 | 0.18 | 0.14 |
| Others | 0.91 | 2.22 | 3.32 | 3.44 | 4.34 | 5.46 |

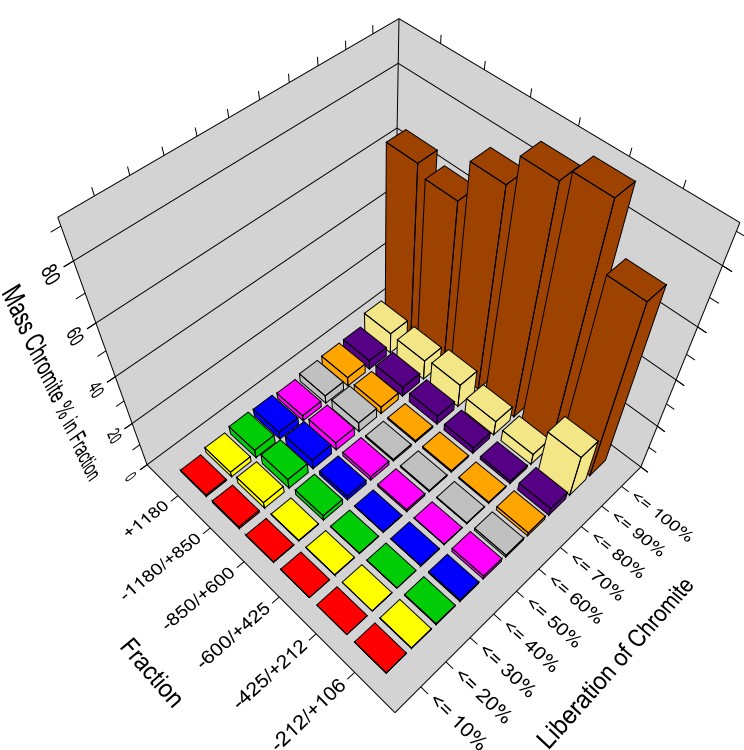

**Figure 6.** Plot of the mass percentages of chromite relative to the level of liberation of chrome-rich spinel grains by size.

### 4.7. Comparison of EPMA and QEMSCAN Results

A comparison of data obtained from two different electron-beam-based instruments showed that both gave similar results, each producing mineral classified images (Figures 4 and 5) that provided valuable quantifiable information on mineralogical and textural relationships, as well as modal abundances for the various phases that were present. Indeed, a comparison of the modal (area %) data obtained for the −600 +425 μm fraction—the only sample measured using both techniques—indicated that both systems could predict ~89% chrome-rich spinel. Agreement in the gangue mineral phase content was also very good.

Several other technologies besides electron-beam-based techniques can be used to derive mineral maps of ores and particles, e.g., hyperspectral imaging and standard RGB color imaging. The selection of the most appropriate technique will depend on the ore characteristics and the intended use of the data. In the case of the chromite ore studied here, both the EPMA and QEMSCAN techniques were suitable for the acquisition of images representative of the mineralogical and textural complexity of the ore. This may not always be the case, as Pownceby et al. [10] examined mineralogical problems which may specifically require the use of an electron microprobe, while numerous other studies have explored the applications of QEMSCAN (and the similar MLA and TESCAN) techniques [14,15]. Nonetheless, as shown in this study, neither technique should be viewed in isolation as they have the potential to offer complementary information. It is also recommended that they are used in association with conventional techniques, such as optical microscopy and other analytical techniques such as XRD and XRF, especially when characterizing highly complex ores.

### 4.8. Implication for Processing Chromite Ore

The mineralogical characterization data indicated that the silica in the sample was mainly present in discrete grains, but lesser amounts were present as silicate–spinel (Cr) composite grains, as free particles, or on the edges of chrome-rich spinel grains. The spinel

grains were found to be >80% liberated. Based on these data, to beneficiate the ore, stage crushing of the ore to a finer size should liberate the chrome-rich spinel grains from the silicate gangue material. Simple gravity concentration could then be used to upgrade the chrome-rich spinel to the required specification.

Results from such a test program on the ore were reported by McCallum at al. [16]. In the study, wet gravity concentration using sized fractions (−600 +106 μm size range) from the original −2 mm ore, as well as ore that had been crushed to pass 600 μm, showed the following:

- the $Cr_2O_3$ grade at 70% $Cr_2O_3$ recovery increased from ~44% $Cr_2O_3$ in the primary crush to 46% $Cr_2O_3$ (the target-grade specification) in the secondary sample.
- using a cut-off level of 1% $SiO_2$, crushing increased the cumulative $Cr_2O_3$ recovery from ~50% in the primary material to ~63% in the secondary sample in the same size range of −600 +106 μm. The extent to which additional crushing could lead to improved chrome-rich spinel recoveries at the required chromia- and silica-grade target specifications was not determined.

These beneficiation results demonstrate that prior to the processing test work being conducted, a thorough understanding of the chemistry, mineralogy, and degree of liberation of the test sample promotes efficient and targeted processing research.

## 5. Summary

A mineralogical characterization of a South African chromite ore was undertaken using bulk assays (XRF), X-ray diffraction, optical SEM, and automated electron beam microanalysis techniques to provide input into the design of a beneficiation treatment process to upgrade the chromite to a marketable metallurgical-grade product.

The bulk material assayed 40.5% $Cr_2O_3$, with the major impurities being $Al_2O_3$ (13.2%), MgO (12.1%), and $SiO_2$ (7.5%). Qualitative mineral phase analysis by XRD indicated that the sample mineralogy was dominated by the ferrian–magnesiochromite and aluminum–chromite spinels, and with enstatite, hematite, and anorthite-rich plagioclase as minor phases. EPMA mapping and quantitative mineral phase analysis confirmed that the sample mineralogy was dominated by a chrome-rich spinel phase with average chemical compositions (in wt.%) of $Cr_2O_3$—47.8, FeO—26.0, $Al_2O_3$—15.4, and MgO—11.0. Contaminant phases included siliceous minerals enstatite, anorthite-rich plagioclase, Cr-rich diopside (containing 1–2 wt.% $Cr_2O_3$), and phlogopite mica.

QEMSCAN analysis of sized fractions indicated that (a) most of the silicate species were present in the +850 μm fractions, (b) the chrome-rich spinel was >80% liberated in all fractions, and (c) the most common mineral association for chromite was with enstatite. Excellent agreement was achieved between both automated electron beam techniques.

Based on the results, preliminary upgrading test work confirmed that stage crushing to liberate the chrome-rich spinel, followed by wet gravity concentration, could produce a chemical–metallurgical-grade chromite assaying >46% $Cr_2O_3$ and <1% $SiO_2$.

**Author Contributions:** Conceptualization, D.A.M., M.I.P. and W.J.B.; methodology, D.A.M. and M.I.P.; formal analysis, M.I.P. and D.A.M.; writing—original draft preparation, M.I.P.; writing—review and editing, M.I.P., D.A.M. and W.J.B. All authors have read and agreed to the published version of the manuscript.

**Funding:** This research received no external funding.

**Acknowledgments:** The authors gratefully acknowledge the assistance of the Analytical Services and Electron Microscopy Groups at CSIRO Mineral Resources. We thank Nick Wilson and Aaron Torpy of the Microbeam Characterization Team (Clayton) for the EPMA mapping and quantitative analyses and Chi Ly (CSIRO Mineral Resources—Waterford) for the QEMSCAN analyses.

**Conflicts of Interest:** The authors declare no conflict of interest.

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
