# Peer review of "Automated and Quantitative Mineralogy Applied to Chromite Ore Characterization and Beneficiation"

_minerals, doi:10.3390/min13030440_

Round 1
Reviewer 1 Report
The paper is not novel enough for publication. It is a routine procedure to determine the properties of the chromite ores before any beneficiation process such as wet gravity methods. For the characterization, the methods applied here are common and widely used methods therefore I do not see any beneficiation for the current literature. Therefore, I recommend that a shorthened version of the paper can be re-submitted as a case study.
Reviewer 2 Report
The manuscript systematically analyzes the process mineralogical characteristics of chromite, which has certain reading interest and value. There are several questions that need the author's attention:
1. How is the mineral composition of enstatite, plagioclase and diopside determined in Fig. 2 only by scanning electron microscope without electron probe?
2. Why is the sum of chemical composition in each mineral in Table 3 not 100%?
3. Why hasn't the mineral composition content of - 105 microns in Table 5 been counted?
4. How is the degree of dissociation determined? Is there a specific calculation equation? Please add in the method description.
5. The ore is broken to - 2mm and analyzed by various means. For the degree of dissolution, what is the significance of studying the degree of dissolution of - 2mm ore- Does the 2mm ore meet the requirements of beneficiation granularity?
Reviewer 3 Report
Authors present results of a creditable and multidisciplinary characterization study of chrome ore. They have used several distinct techniques including XRF assays, XRD and micro probe and scanning electron microscopy-based techniques. All the used techniques represent established technologies, but the merit of this article is to combine results from different sources and use that data to validate the results

Round 2
Reviewer 1 Report
The paper is well studied and experimented on the characterization of the chromite ore taken from a specific chromite mine. The determining/analyzing techniques used here are new and looking promising for the chromite processing industry, but I do repeat my previous conclusion that the paper has no novelty in its current form.
The reasons: i. there is no direct guidance for the processing methods chosen/applied to have beneficiation from data obtained here, ii. there is no specific information here that which wet processing method i.e. shaking tables, spirals, etc. was used for the beneficiation, iii. for the readers, it is too difficult to evaluate the results given here for beneficiation purposes such as optimizing the operation parameters or designing the process flowsheet, etc.
Therefore, the paper should have more info on the beneficiation section showing that these findings on the characterization of the ore have a significant value for the wet processing method applied as mentioned as the aim of the paper. Then, the paper has novelty for the current literature. I do recommend that the paper can be redesigned before being re-submit.
